# Low-Resource Preference Adaptation of LLMs via Activation-Based Label Propagation

## Abstract

Adapting large language models to user-specific preferences is often constrained by the cost of human annotation, making preference optimisation impractical in low-resource settings where preferences cannot be reliably labelled by LLMs themselves, e.g., due to cultural, subjective, or personalised contexts. In this paper, we investigate how language models encode preference information in their intermediate representations, finding that activations from chosen and rejected responses form distinct clusters across layers, even in pretrained models. Exploiting this structure, we propose training a lightweight linear probe on a few labelled preference pairs ($\leq 500$) and using it to annotate large unlabelled datasets (50K+) for downstream preference optimisation. We systematically evaluate this approach across different datasets, preference optimisation methods and model scales (1B-14B). Models trained with probe-generated labels consistently outperform both supervised fine-tuned baselines and models trained on random labels. Compared to the original labels, performance is more mixed and depends on the robustness of the optimisation method. Our findings democratise preference optimisation for low-resource settings, enabling effective adaptation in domains where human annotation budgets are severely constrained, including underrepresented and marginalised communities whose preferences have historically been underserved by large-scale models.

## 1. Introduction

Preference optimisation has emerged as a critical technique for aligning language models with human values and expec-

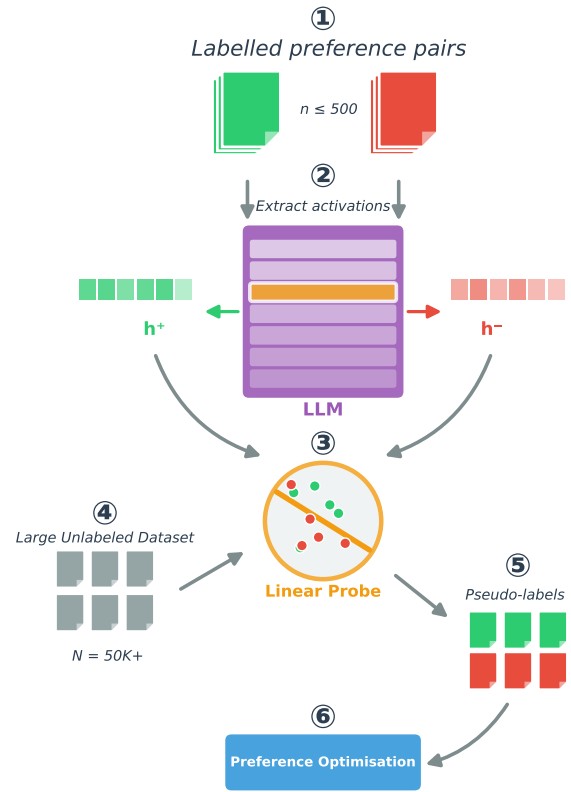

*Figure 1.* Illustration of our proposed pipeline. We start with (1) a small dataset with labelled preference pairs. Using the difference in activations of an LLM (2), we train a linear probe (3). We then use the probe to label a large corpus (4, 5) and do preference optimisation (6).

tations ([Ouyang et al., 2022](); [Rafailov et al., 2023]()). Methods such as Direct Preference Optimization (DPO) ([Rafailov et al., 2023]()), IPO ($\Psi$PO with identity mapping) ([Gheshlaghi Azar et al., 2024]()), and their variants enable models to learn from human judgments about which responses are preferable. However, these methods typically require large datasets of labelled preference pairs, with annotation costs scaling linearly with dataset size.

This requirement poses challenges for adapting models to specific user populations. Different communities, organisations, and cultural contexts exhibit distinct preferences

---

[1]Anonymous Institution, Anonymous City, Anonymous Region, Anonymous Country. Correspondence to: Anonymous Author <anon.email@domain.com>.

Preliminary work. Under review by the International Conference on Machine Learning (ICML). Do not distribute.

(Kirk et al., 2024; Sorensen et al., 2024)—what constitutes a helpful or appropriate response varies substantially across groups. Obtaining large-scale annotations from each target population is often impractical, creating a barrier to personalised alignment.

A natural question arises: can we reduce the annotation burden while still achieving effective preference adaptation? One approach is to use an LLM (e.g., GPT-4 or newer) as an external judge (Zheng et al., 2023), but these encode their own preference biases rather than those of the target population (Li et al., 2025). Reward model distillation requires substantial labelled data for training. In this work, we propose a methodology to capture population-specific preferences with minimal supervision.

First, we investigate how LLMs encode preference-relevant information in their activations. Our results show that when processing chosen versus rejected responses, the resulting representations occupy different regions of activation space, forming clusters with distinct centroids even before preference optimisation (Figure 2). When using 'traditional' preference optimisation datasets (i.e., datasets we expect to be similar to those used internally to train these models), this separation becomes more pronounced after supervised fine-tuning and preference optimisation, suggesting that instruction-following training implicitly organises representations according to response quality.

Building on this observation, we propose a simple pipeline (illustrated in Figure 1) for label-efficient preference adaptation:

1. Collect a small set of labelled preference pairs from the target population ($n \approx$ 100-500).

2. Train a linear probe (or similar small model) on model activations to distinguish chosen from rejected responses.

3. Apply the probe to label a large corpus of unlabelled response pairs.

4. Run standard preference optimisation using the probe-generated labels.

We systematically evaluate this approach and find:

- **Probe accuracy matches inter-annotator agreement.** Linear probes achieve 60-80% accuracy using only 100-500 training samples, comparable to the inter-annotator agreement observed in preference datasets.

- **Noisy probe labels are enough for Preference Optimisation.** Models trained on probe labels consistently outperform both randomly labelled datasets and SFT baselines.

- **Preference optimisation methods differ drastically in robustness.** IPO trained on probe labels can exceed fully-supervised performance, while DPO degrades substantially at various model scales. We also test label smoothing in DPO, which helps significantly reduce this effect.

Our contributions are threefold: (1) we document the geometric structure of preference representations in LLM activations; (2) we provide a systematic analysis of preference optimisation robustness under noisy labels; and (3) we demonstrate a practical method for preference adaptation requiring $100\times$ fewer human annotations.

## 2. Related Work

### 2.1. Preference Optimisation

Reinforcement Learning from Human Feedback (RLHF) (Christiano et al., 2017; Ouyang et al., 2022) aligns language models with human preferences by training a reward model and optimising against it via reinforcement learning. Direct Preference Optimisation (DPO) (Rafailov et al., 2023) simplifies this by reparameterising the reward model as an implicit function of the policy, enabling direct optimisation on preference pairs. Subsequent work has proposed variants addressing limitations of DPO: IPO (Gheshlaghi Azar et al., 2024) avoids overfitting via a different loss formulation, KTO (Ethayarajh et al., 2024) operates on unpaired examples using Kahneman-Tversky's model of utility (Tversky & Kahneman, 1992), and CPO (Xu et al., 2024) incorporates contrastive objectives. Our work studies how these methods respond to label noise, revealing substantial differences in robustness.

### 2.2. Label-Efficient Alignment

Prior work has explored reducing annotation requirements for alignment. Constitutional AI (Bai et al., 2022b) uses model self-critique, while RLAIF (Lee et al., 2024) employs LLM judges for labelling. However, these approaches inherit the preferences of the labelling model rather than the target population. Reward model distillation (Fisch et al., 2025; Askell et al., 2021) transfers preferences but still requires substantial seed data. Recent work on weak-to-strong generalisation (Burns et al., 2024) studies supervision from less capable models. Our approach differs by exploiting structure in the model's own representations.

### 2.3. Linear Probing in LLMs

Linear probes have been used extensively to study representations in language models, revealing that models encode syntactic (Hewitt & Manning, 2019), semantic (Tenney et al., 2019), and factual (Meng et al., 2022) information

in linearly accessible ways. Recent work has applied probing to detect sentiment (Tigges et al., 2024) hallucinations (Azaria & Mitchell, 2023), truthfulness (Marks & Tegmark, 2024), refusal behaviour (Arditi et al., 2024), model uncertainty (Wang et al., 2025; Dakhmouche et al., 2025) and answer accuracy (Cencerrado et al., 2025).

We extend this line of work by showing that preference information is similarly encoded and can be exploited for practical label propagation. To the best of our knowledge, we are the first to use probes for label propagation at scale for preference optimisation.

## 3. Preference Geometry in Activations

We begin by characterising how preference information is organised in language model representations.

### 3.1. Experimental Setup

We analyse activations from different models of Llama 3.2 (Llama Team, 2024), Gemma 3 (Team, 2025a), and Qwen 3 (Team, 2025c) at various training stages: pretrained, or after SFT and Preference Optimisation (PO). For each stage, we process preference pairs from HH-RLHF (Bai et al., 2022a), Ultrafeedback (Cui et al., 2023), or Nectar (Zhu et al., 2024). On top of these 'standard' datasets, we also analyse PRISM (Kirk et al., 2024) as an example of culturally diverse preferences. For each sample, we extract the activations of positive and negative samples. The activations for a single sequence are then aggregated by either taking: (i) the mean of the completion tokens, (ii) only the token with maximum activations, or (iii) only the activations for the final token in the sequence. We also vary the number of training samples from 10 to 500.

### 3.2. Cluster Structure

We visualise activations from chosen and rejected responses using t-SNE (van der Maaten & Hinton, 2008) and PCA (Pearson, 1901), noticing that, while chosen and rejected clusters mostly overlap, their centroids are quite distinct. This separation is present even in pretrained models but becomes more pronounced (up to 40% more) after SFT and PO. To show this separation is not caused by confounding factors, we also plotted (and report in the Appendix) activations with labels randomly swapped. In this latter case, the clusters completely overlap and focus around the same centroid.

We show in Figure 2 one of the most prominent examples, obtained with Qwen 3 0.6B and the Nectar dataset. While our observations stand across models, families and datasets. We found that the latter has the biggest effect on sample separation, where, e.g., HH-RLHF shows a less pronounced difference (although still present).

### 3.3. Layer-wise Analysis

To quantify separability, we train linear probes at each layer and measure classification accuracy on held-out data. Figure 3 shows probe accuracy across layers for the same setting as Figure 2.

We find that probe accuracy peaks in the middle-to-late layers, consistent with findings on other representation probing tasks (Gurnee & Tegmark, 2024). We also find that the accuracy of pretrained and PO models is mostly similar, with some families of models showing slightly higher accuracy for pretrained (e.g., Llama 3.2), while others showing slightly higher for PO (Gemma 3, Qwen 3). Similar to before, we find that the dataset has the biggest influence on probe accuracy, with Nectar peaking above 80% while HH-RLHF plateaus around 62-65%.

Note that, while probe accuracy of 62% may appear low, preference annotation can be quite noisy. In HH-RLHF, the reported inter-annotator agreement is approximately 63%, indicating that our probe approaches the ceiling of human consistency. This suggests the probe may capture a genuine preference signal rather than merely failing to learn the task.

### 3.4. Variability

Varying the aggregation method, i.e., mean of activations vs. last token vs. max, does not affect clustering nor probe performance. However, an important aspect of computing the mean is that it should be calculated solely over the completion, excluding the prompt. Averaging over the entire input leads to clusters that are no longer distinguishable. We also found consistency across model families, sizes and datasets, except for small fluctuations in performance between probes trained on 10 to 500 samples.

### 3.5. Culturally Diverse Preferences

The main application of our findings is in low-resource scenarios and culturally specific settings, where data is too scarce, we analyse the activations of PRISM (Kirk et al., 2024). PRISM is a preference dataset collected from a very diverse range of populations. Each sample is paired with the cultural background of the annotator. We begin by grouping conversations in PRISM by user location and age group, and we discover similar results in terms of clustering and probe performance to that of 'standard' preference datasets. A key difference here is that, while the pretrained model shows distinct centroids for chosen and rejected samples, this distinction is much reduced or not present *at all* in the default SFT/PO version of the model (see Figure 4). This strengthens the hypothesis that models aligned to a particular set of preferences are unable to distinguish between a *different* set of preferences and are thus unfit to be used as judges or annotators.

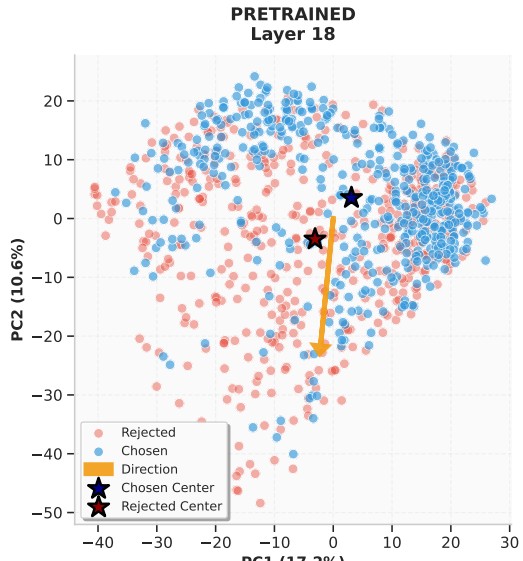
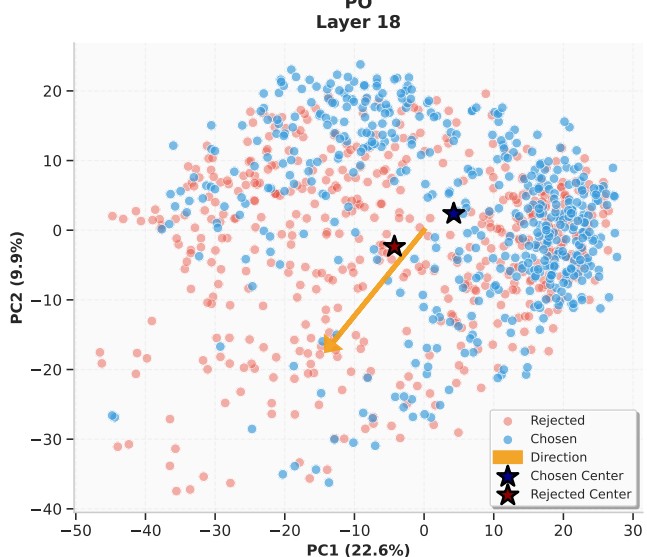

*Figure 2.* Activations from chosen (blue) and rejected (orange) responses form separable clusters using Qwen 3 0.6B and Nectar. The distance between centroids increases in PO compared to pretrained.

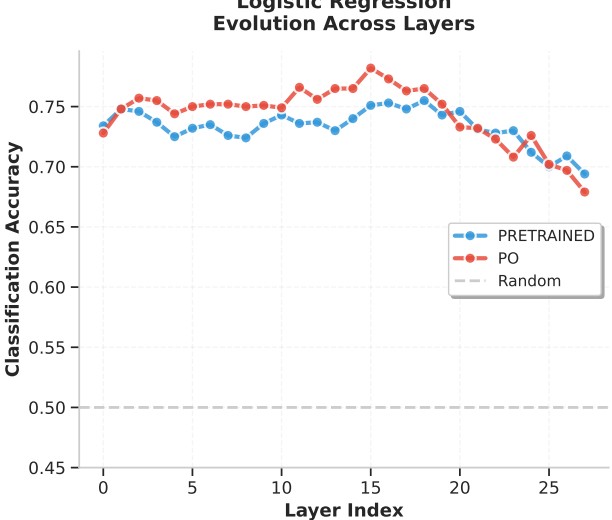

*Figure 3.* Linear probe accuracy across layers on Qwen 3 0.6B. Accuracy peaks around layer 15, and the PO model shows higher accuracy consistently across most layers.

# 4. Method

We describe our pipeline for label-efficient preference adaptation assuming access to:

- A language model $\pi_\theta$ fine-tuned for instruction following (SFT model)

- A small labelled dataset $\mathcal{D}_L = \{(x_i, y_i^+, y_i^-)\}_{i=1}^n$ with $n \approx 10\text{-}500$

- A large unlabelled dataset $\mathcal{D}_U = \{(x_j, y_j^a, y_j^b)\}_{j=1}^N$ with $N \gg n$

The goal is to adapt $\pi_\theta$ to the preferences represented in $\mathcal{D}_L$ by leveraging $\mathcal{D}_U$.

## 4.1. Probe Training

For each labelled example $(x, y^+, y^-)$, we extract from $\pi_\theta$ activations at layer $\ell$ and select those relative to the last token in the sequence:

$$h^+ = \text{Activation}_\ell(\pi_\theta, [x; y^+]) \tag{1}$$

$$h^- = \text{Activation}_\ell(\pi_\theta, [x; y^-]) \tag{2}$$

We train a linear probe $f_\phi(h) = \sigma(Wh + b)$ to predict preference:

$$\mathcal{L}_{\text{probe}} = -\sum_i \left[ \log f_\phi(h_i^+) + \log(1 - f_\phi(h_i^-)) \right] \tag{3}$$

Here, the choice of $\ell$ can be either made through a validation set or with an 'informed' guess. Based on the analysis in Section 3, we believe the best layers to be those starting from the middle of the model, e.g., for a model with $L$ layers, layers $\lfloor L/2 \rfloor$ to $\lfloor 2L/3 \rfloor$.

## 4.2. Label Propagation

For each unlabelled pair $(x, y^a, y^b) \in \mathcal{D}_U$, we compute:

$$p^a = f_\phi(\text{Activation}_\ell(\pi_\theta, [x; y^a])) \tag{4}$$

$$p^b = f_\phi(\text{Activation}_\ell(\pi_\theta, [x; y^b])) \tag{5}$$

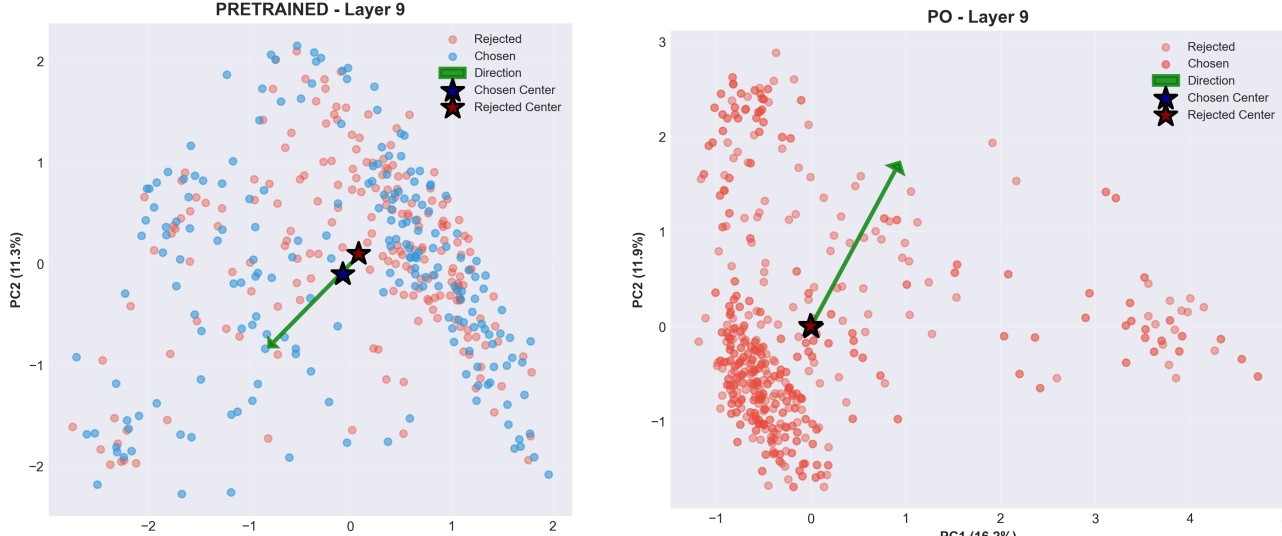

*Figure 4.* Activations from chosen (blue) and rejected (orange) responses form separable clusters using Llama 3.2 1B and PRISM. Distance between centroids *decreases* in PO compared to pretrained.

We assign labels based on probe predictions:

$$(y^+, y^-) = \begin{cases} (y^a, y^b) & \text{if } p^a > p^b \\ (y^b, y^a) & \text{otherwise} \end{cases} \quad (6)$$

### 4.3. Preference Optimisation

We apply standard preference optimisation methods (DPO, IPO, KTO, CPO) on the probe-labelled dataset. The key question is how each method responds to the noise introduced by imperfect probe predictions.

## 5. Experiments

We conduct several experiments to validate the effectiveness of models trained on probe labels.[1]

### 5.1. Experimental Setup

**Models.** We experiment with Llama 3, Gemma 3, Qwen 3 at varying scales from 1B to 14B parameters.

**Probe training.** Unless otherwise stated, the probe is trained on 500 examples, with activations from the three layers following the middle one (e.g., for a 16-layer model, we use layer 7, 8 and 9). The probe is then used to re-label (up to) 50K examples.

**Datasets.** We begin with a common SFT training using the instruction tuning dataset 'Alpaca' (Taori et al., 2023). This

---

[1]Code for all the analysis and experiments is available in the supplementary material.

dataset is useful to provide the model with *some* instruction following behaviour without focusing on the harmlessness-helpfulness that is generally injected in the preference optimisation phase. After SFT, we proceed with preference optimisation, where we experiment with HH-RLHF (Bai et al., 2022a), Ultrafeedback Binarized (Cui et al., 2023), and Nectar (Zhu et al., 2024). For each dataset/training run, we hold out 10-500 examples for probe training, reserving ∼50K for preference optimisation and evaluation.

**Methods.** We compare DPO (Rafailov et al., 2023), IPO (Gheshlaghi Azar et al., 2024), KTO (Ethayarajh et al., 2024), and CPO (Xu et al., 2024). For all the experiments, we use LoRA (Hu et al., 2022) to reduce computational demand.

**Evaluation.** We evaluate using LLM-as-a-judge on 500 samples from the original dataset using 'Qwen 2.5 14B Instruct' as a judge (Team, 2025b). While we also tested other judges, we found this to be a good trade-off between performance, consistency and size. Our initial tests utilised Llama 3.2 1B-3B (Instruct), however, both yielded high variability and tie rates. After testing smaller models from the Qwen 2.5 family, we selected the 14B variant as the most reliable. Nevertheless, we report high consistency even among the different judges, where most judges would, on average, select the same winner model.

**Baselines.**

- **Original labels**: Standard preference optimisation with original annotations.

- **Random labels**: Uniform random assignment of chosen/rejected (i.e., half the samples will have swapped labels).

- **SFT:** To verify the model actually improves compared to before training.

### 5.2. Main Results

Table 1 and Figure 5 present our main comparison across methods and model scales. As HH-RLHF showed the worst probe performance in Section 3, we dedicate most of our experiment to this dataset, confident that any results would most likely generalise to the other datasets as well.

**Baselines.** Models trained using our labelling method consistently outperform our baselines, showing improvement over the SFT checkpoint and over PO models trained with random labels.

**IPO exhibits high robustness to noisy labels.** At both small and big scales, IPO with probe labels is very close and sometimes outperforms models trained on the original dataset.

**DPO degrades substantially.** On the other hand, DPO shows the largest gap between original and probe labels, again, regardless of scale.

**Other methods are competitive.** The other methods, CPO and KTO, show competitiveness between probe-labelled and original-labelled. In most cases, the original-labelled still exhibits better performance, but with very small margins.

**Some models degenerate.** Perhaps unsurprisingly, we find that the smallest model tested, 'Qwen 3 0.6B', degenerates quite easily both with original but also probe-labelled examples. More interestingly, we find that IPO tends to yield degenerated models when trained with *original* labels but not with the probe's. On the opposite direction, DPO tends to yield degenerated models when trained with probe labels but not with original's.

### 5.3. Effect of Label Smoothing

Label smoothing has been proposed to improve DPO robustness (Mitchell, 2023). Table 3 compares different smoothing values.

Higher smoothing substantially improves robustness, increasing the win-tie rate from 37% to 67.2%. This aligns with the interpretation that DPO overfits to noisy labels.

### 5.4. Probe Training Set Size

Figure 6 shows performance as a function of probe training set size. We test the following configurations: $n \in \{10, 50, 100, 250, 500, 1000\}$. We plot the win-rate of the probe-trained model against the SFT model.

While the plot shows almost monotonically increasing win-tie rate, the biggest increase in performance happens with 250 training samples until 500. After this boundary, increasing training size yields diminishing results (while still improving overall performance).

### 5.5. Other Datasets

We present the results for the UltraFeedback and Nectar datasets in Tables 4 and 5, respectively. Due to computational demand, we only analyse one model per family. Nevertheless, the results are very similar to those from HH-RLHF: models trained on probe labels often outperform randomly labelled datasets and, only occasionally, the models trained with the original dataset.

### 5.6. Computational Overhead

Probe training is negligible: it requires $n$ forward passes to extract activations (where $n \leq 500$) followed by fitting a linear classifier, completing in under a minute on a single GPU. Label propagation (on 50K+ samples) in our current implementation adds approximately 70% overhead to preference optimisation training time, as we perform labelling as a separate preprocessing step. However, this overhead can be eliminated entirely by fusing label propagation with preference optimisation. Specifically, during each training iteration, one can perform a forward pass on an unlabelled pair $(y^a, y^b)$, extract activations to obtain probe predictions, assign labels accordingly, and then compute the preference optimisation loss using the same forward pass outputs before backpropagating. This fused approach adds only the cost of a single linear probe evaluation per batch—negligible compared to the LLM forward and backward passes. We provide a sketch of this algorithm in the Appendix.

## 6. Discussion

**Practical recommendations.** From our systematic evaluation, we derive the following recommendations for practitioners. First, prefer IPO over DPO when using probe-generated labels: IPO consistently tolerates label noise across all model scales, sometimes matching or exceeding fully-supervised performance, while DPO degrades substantially (Table 1). If DPO is required (e.g., for compatibility with existing pipelines), use label smoothing $\geq 0.25$, which improves win-tie rates from 37% to over 65% (Table 3). Second, collect at least 250 labelled preference pairs for probe

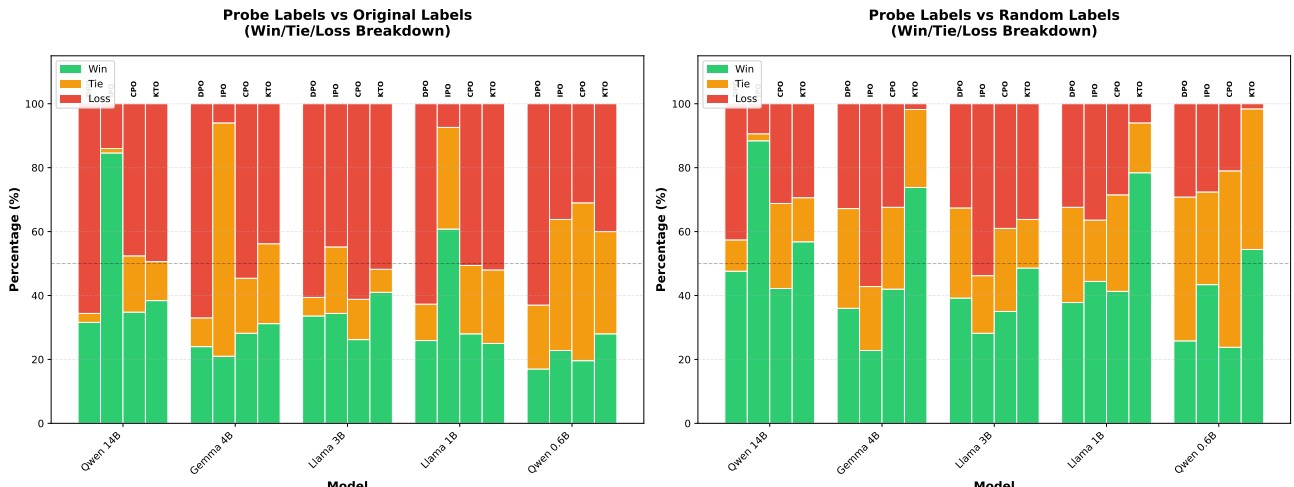

*Figure 5.* Illustration of the results from Table 1.

*Table 1.* Probe labelled models *win+tie* rate in percentage % (tie rate also in parentheses) compared to original-labelled models or randomly labelled model after full training on HH-RLHF. Higher indicates probe labels perform better. We highlight in **bold** the times the probe-labelled model has a higher win-rate than its baseline (i.e., excluding ties). We mark with a '*' the runs where one or both models occasionally express degenerated outputs.

| Family | Size | Original | | | | Random | | | |
|--------|------|------|------|------|------|------|------|------|------|
| | | DPO | IPO | CPO | KTO | DPO | IPO | CPO | KTO |
| Qwen 3 | 14B | 34.4 (2.8) | **86 (1.4)** | 52.4 (17.6) | 50.6 (12.2) | **57.4 (9.8)** | **90.6 (2.2)** | 68.8 (26.6) | 70.6 (13.8) |
| Gemma 3 | 4B | 33 (9) | **94* (73)** | 45.4 (17.2) | 56.2 (25) | 67.2 (31.2) | 42.8* (20) | 67.6 (25.6) | 98.2 (24.4) |
| Llama 3.2 | 3B | 39.4 (5.8) | 55.2 (20.8) | 38.8 (12.6) | 48.2 (7.2) | 67.4 (28.2) | 46.2 (18) | 61 (26) | 63.8 (15.2) |
| Llama 3.2 | 1B | 37.3 (11.4) | **92.6 (31.8)** | 49.4 (21.4) | 48 (23) | 67.6 (29.8) | 63.6 (19.2) | 71.5 (30.2) | 94 (15.6) |
| Qwen 3 | 0.6B | 37 (20) | 63.8* (41) | 69* (49.4) | 60* (32) | 70.8 (45) | **72.4* (29)** | 79 (55.2) | 98.4 (44) |

*Table 2.* Probe labelled models *win+tie* rate in percentage % (tie rate also in parentheses) compared to Supervised Fine-Tuned (SFT) checkpoint on HH-RLHF. Higher indicates probe labels perform better. We highlight in **bold** the times the probe-labelled model has a higher win-rate than its baseline (i.e., excluding ties).

| Family | Size | SFT | | | |
|--------|------|------|------|------|------|
| | | DPO | IPO | CPO | KTO |
| Qwen 3 | 14B | **80.9 (9.80)** | **91.6 (0.8)** | 52.6 (16.8) | **65 (10.4)** |
| Gemma 3 | 4B | **69.8 (30)** | 32.4* (11) | 60.2 (21.8) | **58.6 (15)** |
| Llama 3.2 | 3B | **70.4 (28.2)** | 50.6 (21.2) | 55.2 (20.6) | **62.4 (18.2)** |
| Llama 3.2 | 1B | **73.8 (33.4)** | **63.8 (24.6)** | **61.6 (22.6)** | **62.8 (20.2)** |
| Qwen 3 | 0.6B | **77.8 (40.6)** | **73 (30.6)** | **74.4 (44.6)** | **63.6 (38.6)** |

*Table 3.* Effect of label smoothing on DPO with probe labels (Qwen3 0.6B, HH-RLHF). Win rate (%) against original labels. We highlight in **bold** the best results for the probe.

| Label Smoothing | Win | Lose | Tie |
|-----------------|------|------|------|
| 0.0 | 17 | 63 | 20 |
| 0.1 | 18.4 | 57.2 | 24.4 |
| 0.25 | **25.4** | 40 | 34.6 |
| 0.4 | 24.6 | **32.8** | **42.6** |

training; performance improves most steeply between 100-500 samples, with diminishing returns beyond this range (Figure 6). Third, extract activations from middle-to-late layers; for a model with $L$ layers, layers $\lfloor L/2 \rfloor$ to $\lfloor 2L/3 \rfloor$ consistently yield the highest probe accuracy across model families (Figure 3). Fourth, when aggregating token activations, use the mean over completion tokens only, excluding the prompt; including prompt tokens causes chosen and rejected clusters to become indistinguishable. Finally, if probe accuracy on held-out data falls below 55%, consider

collecting additional labelled examples or verifying that the target preferences are sufficiently distinct from random.

**Generalisability to specific preferences.** Throughout this work, we mostly relied on 'standard' preference datasets. However, the primary application of our method is in population-specific contexts where obtaining large labelled datasets is impractical or too expensive. Datasets such as PRISM (Kirk et al., 2024) offer a set of diverse preferences for training; however, comparing models trained on this dataset can be highly unreliable without access to the original population. Our pipeline relies on LLM-as-a-judge,

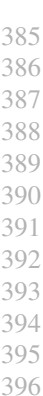

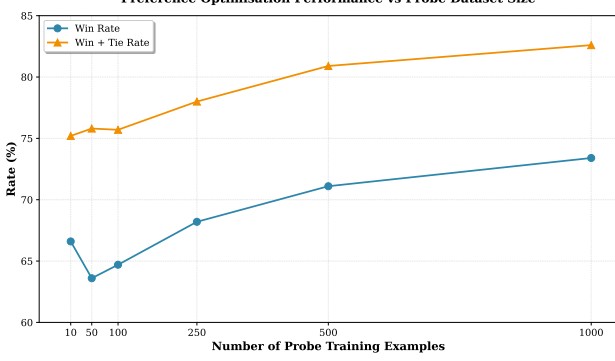

*Figure 6.* Effect of probe training set size on downstream preference optimisation performance. Reported is the win-rate against the SFT model. Performance starts to rise after 100 samples.

*Table 4.* Probe labelled models *win+tie* rate in percentage % (tie rate also in parentheses) on UltraFeedback dataset. Baselines are Original (O) and Random (R). Higher indicates probe labels perform better. We highlight in **bold** the times the probe-labelled model has a higher win-rate than its baseline (i.e., excluding ties).

| Model | Vs | DPO | IPO | CPO | KTO |
|-------|----|-----|-----|-----|-----|
| Qwen 0.6B | O | 45.8 (21.2) | **86\* (33.4)** | 48.2 (20.4) | 23.6\* (6.8) |
|  | R | **70.6 (33.2)** | **68.6\* (30.2)** | 61.2 (23.2) | **95\* (23.6)** |
| Llama 1B | O | 49.8 (14.2) | **93.6 (31.4)** | 44 (15.2) | 39.6 (7.8) |
|  | R | **65.6 (30.2)** | 51 (7.6) | 60.2 (23) | **77.6\* (4.2)** |
| Gemma 4B | O | 30.4 (11) | 45.8 (3.2) | 34.6 (3.4) | 31.6 (4) |
|  | R | 57.2 (20.4) | **92.6 (11.2)** | **56 (4.8)** | **78.2 (6)** |

which has similarly been shown to be unreliable for 'personalised' preferences e.g., Dong et al. (2024) use PRISM and report very low ground truth agreement for 'personalised' LLM judges. Nevertheless, our analysis in Section 3 on PRISM reveals similar structures and probe performance compared to 'standard' datasets. We thus hypothesise that this would transfer to preference optimisation as well, with similar remarks on robustness and stability.

## 7. Conclusion

We demonstrated that language models encode preference information in their activations as distinct clusters. Exploiting this structure, we proposed a method for label-efficient preference adaptation: train a linear probe on a few hundred labelled examples, propagate labels to a large unlabelled corpus, and run preference optimisation. Our systematic evaluation revealed substantial differences in method robustness: IPO tolerates noisy labels while DPO degrades unless label smoothing is used. Nevertheless, training using probe labels has shown consistent improvement over SFT models and random labelling. This work enables preference adaptation with $100\times$ fewer annotations, making personalised alignment more accessible, especially in low-resource settings.

*Table 5.* Probe labelled models *win+tie* rate in percentage % (tie rate also in parentheses) on Nectar dataset. Baselines are Original (O) and Random (R). Higher indicates probe labels perform better. We highlight in **bold** the times the probe-labelled model has a higher win-rate than its baseline (i.e., excluding ties).

| Model | VS | DPO | IPO | CPO | KTO |
|-------|----|-----|-----|-----|-----|
| Qwen 0.6B | O | 38.8 (21.6) | **80.6 (31.6)** | 43 (17) | 33.8 (10.4) |
|  | R | **68 (34)** | 64.6 (29) | **61.6 (21.4)** | **94.8\* (27.4)** |
| Llama 1B | O | 55.6 (24.2) | **92.6\* (33.4)** | 53.6 (14) | 33 (13.2) |
|  | R | 65.6 (28.8) | 45 (16.4) | **58.8 (14.6)** | **83 (28)** |
| Gemma 4B | O | 21 (5.8) | **77.4\* (2.2)** | 37.4 (5.8) | 47.4 (4.6) |
|  | R | **68.2 (18)** | **61.6 (3.8)** | **59 (6.6)** | **95.8 (4.8)** |

### 7.1. Limitations

Our work has several limitations that suggest directions for future research. First, our evaluation relies on LLM-as-a-judge, which creates tension with our motivation that LLM judges encode their own preference biases. While we found Qwen 2.5 14B to be consistent and performant on 'standard' preference datasets, human evaluation would provide stronger validation, particularly for culturally-specific preferences. Second, although we demonstrate that PRISM exhibits a similar activation geometry to standard datasets (Section 3.5), we do not evaluate downstream preference optimisation on PRISM because it is infeasible to access the original annotator population for evaluation. Third, our probe accuracy is bounded by the inherent noise in preference datasets: on HH-RLHF, both our probe and human inter-annotator agreement plateau around 63%, suggesting this may be a fundamental ceiling rather than a limitation of our method. Finally, while we demonstrate robustness across three model families, the optimal layer for probe training may vary for architectures substantially different from the decoder-only transformers we studied.

## Impact Statement

This work's goal is to make preference optimisation more accessible in low-resource settings where manual or automatic labelling is infeasible. Our method significantly reduces the annotation burden required for preference optimisation, potentially democratising access to model adaptation for underrepresented and marginalised communities whose preferences have historically been underserved by large-scale models. By enabling effective preference adaptation with as few as 100-500 labelled examples, our approach could allow organisations with limited budgets, minority language communities, and culturally distinct populations to better align language models with their specific values and needs.

On the other hand, the reliance on probe-generated labels also introduces noise that, while manageable for robust optimisation methods, could lead to unpredictable model behaviours if not carefully validated. We encourage practitioners to carefully audit both the seed preference data and

the resulting model outputs when applying this method.

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

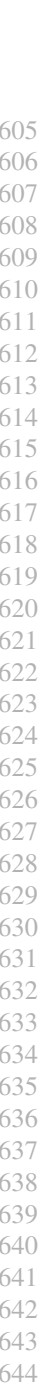

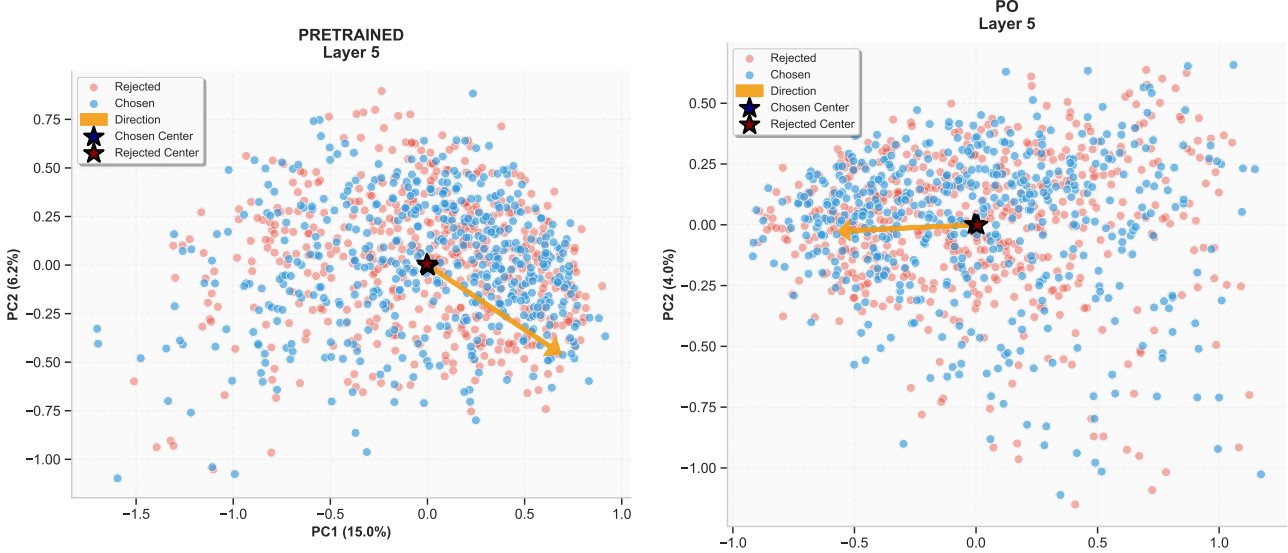

*Figure 7.* **Sanity check:** example clustering with labels randomly swapped. The indistinguishable nature of the clusters and centroids confirms that the difference in activations in Figure 2 is given by the contrastive nature of positive/negative examples.

## A. Additional results

### A.1. Sanity check on clustering

### A.2. Optimised probe inference algorithm

## B. Hyperparameters

*Table 6.* Hyperparameters for preference optimisation methods.

| Hyperparameter | Value |
|---|---|
| Learning rate | 1e-4 (selected from an interval of $[5e-4, 1e-5]$) |
| (Virtual) Batch size | 64 |
| $\beta$ (DPO/IPO) | 0.1 |
| Label smoothing | 0.0 (unless otherwise stated) |
| Max length | None for HH-RLHF, 1024 otherwise |
| Epochs | 1 |
| Infrastructure | A100 for training, A40 for evaluation |

---

**Algorithm 1** Preference Optimisation with Optional Probe Labelling

---

**Require:** Policy $\pi_\theta$, dataset $\mathcal{D} = \{(x_i, y_i^a, y_i^b)\}_{i=1}^N$, PO method $\mathcal{L}_{\text{PO}}$ (e.g., DPO, IPO)
**Require:** *Optional:* trained probe $f_\phi$, extraction layer $\ell$

1: **for** each training step **do**
2:    Sample batch $\mathcal{B} \subset \mathcal{D}$
3:    **for** $(x, y^a, y^b) \in \mathcal{B}$ **do**
4:       $\mathbf{h}^a, \text{logits}^a \leftarrow \text{FORWARD}(\pi_\theta, [x; y^a])$ # Cache activations at layer $\ell$
5:       $\mathbf{h}^b, \text{logits}^b \leftarrow \text{FORWARD}(\pi_\theta, [x; y^b])$
6:
7:       **if** probe $f_\phi$ provided **then**
8:          # Probe labelling (negligible cost)
9:          $p^a \leftarrow f_\phi(\mathbf{h}_\ell^a) \quad p^b \leftarrow f_\phi(\mathbf{h}_\ell^b)$ # Single linear evaluation
10:         $(y^+, y^-) \leftarrow (y^a, y^b)$ if $p^a > p^b$ else $(y^b, y^a)$
11:      **else**
12:         $(y^+, y^-) \leftarrow$ original labels from $\mathcal{D}$
13:      **end if**
14:
15:      Compute $\mathcal{L}_{\text{PO}}(\text{logits}^+, \text{logits}^-; \pi_\theta, \pi_{\text{ref}})$ # Reuse forward pass results
16:   **end for**
17:   Backpropagate and update $\theta$
18: **end for**

---

