# OpenReview forum: "Low-Resource Preference Adaptation of LLMs via Activation-Based Label Propagation"
_ICML.cc/2026/Conference — Submitted to ICML 2026_

### Official Review · Reviewer_WTQ1 · 2026-03-09

**Soundness:** 1
**Presentation:** 3
**Significance:** 2
**Originality:** 2
**Overall Recommendation:** 4
**Confidence:** 4

**Summary:**

In this paper, the authors propose a methodology for performing preference learning in a low-resource environment, assuming that preference labelling is expensive. To this end, they propose using a probe of the middle layers and a small set of labelled preferences (10-500) to learn a preference classifier that can subsequently label more preferences (~50K).
They perform experiments with 5 different models (from the Llama, Gemma, and Qwen families) and test them on the HH-RLHF dataset and also on the PRISM and nectar preference test set.

**Compliance With Llm Reviewing Policy:**

Affirmed.

**Final Justification:**

The authors have addressed all of my concerns; therefore, I raise my score to 4.

**Key Questions For Authors:**

**Q1** Is the data in the classifier evaluation balanced?

**Q2** What does the "original data** in the baselines refer to? (see W5)

**Limitations:**

Yes

**Strengths And Weaknesses:**

# Strengths:
**S1:** The idea is simple to follow, straightforward, and well-explained in the paper
**S2:**  Generally, the problem of low resource preference alignment sounds like an interesting problem that might be underaddressed compared to large-scale preference learning

# Weaknesses:
Unfortunately, this paper has several weaknesses that, in my opinion, need to be addressed first. I hope that the authors don't take this criticism personally, but rather as a way to make a stronger paper.

**W1:** My biggest concern is the experiments and the way they are represented. In particular, I find the win-tie-loss representation and the often-referenced (win-tie) ratio misleading. The tie ratio can be seen as a measure of uncertainty (indicating that it can be either), but in many cases, it is very large (sometimes even larger than the win or loss rate). To this end, I would encourage the authors to test only the Win-Loss ratio (preferably length-controlled as well, e.g., using the Alpaca Eval 2.0 library). With that, you could also add the standard error, a measure of uncertainty for the win-rate. This would make the evaluation of the experiments significantly clearer, as I currently have a hard time determining whether the improvement in their method is actually significant.

**W2:** I am not particularly convinced about the results in Figure 2. While the centroids do not exactly overlap, this does not mean the data is well separated (at least, if we look at the data itself, I am not sure I could cluster it). To determine whether the data in 2D space are well-separated, I would try to introduce a statistical test. An idea could be to reduce all to PC1, then run a t-test on the centroids (means) of the two clusters to see if they differ significantly

**W3:** The models are all newer than the dataset that the authors are testing on. I would therefore be a bit sceptical that this data has not already been seen in the pretraining of all these models.

**W4:** The classifier in Figure 3 looks pretty good, but I would further appreciate it if confidence intervals were given. Moreover, just to double-check, can we confirm that the data is 50/50 balanced in terms of preferences when evaluating it? Otherwise, accuracy might be a misleading metric

**W5**: The baselines are often an off-the-shelf preference optimisation method (DPO, IPO, etc.), compared on the original data and the probe-labelled data. What does the "original data" stand for here? Is that the 10-500 labelled points, or is this the whole dataset (50K) if all labels were available? I feel like both of these would be interesting baselines to compare against.

---

> ### Author Rebuttal · Authors · 2026-03-30
>
> We thank Reviewer WTQ1 for the constructive feedback and the recognition that our problem setting is interesting and underaddressed. We address each concern below, and we believe the new experiments and clarifications resolve the key issues raised.
>
> ## W1: Win-Tie-Loss representation and significance
> We agree that win+tie ratio may be difficult to disentangle; for this, we also refer to Figure 5 as a good visualisation of our results.
> The main reason for the inclusion of ties is to show that with only 500 instances, we are able to tie with a model that needs 100× more data (50K instances). On strengthening our evaluation, we also refer to the replies to Reviewer UCyP and f91J.
>
> ## W2: Statistical validation of cluster separation
> Following the reviewer's suggestion, we conducted a systematic battery of statistical tests across 9 settings (3 datasets × 3 model families), using the same configurations as Section 3.1.
>
> Specifically, we project activations onto PC1 and apply a two-sample t-test (or Mann-Whitney U when normality assumptions are violated). In 5 out of 9 settings, we observe p << 0.001; in 3 of the remaining 4, we still find p < 0.05. Only one combination fails to reach significance at the 0.05 level.
>
> To complement this, we compute Cohen's d on the full-dimensional activation space (not just PC1), obtaining values ranging from 2.63 to 3.75 across all settings. By standard conventions, d > 0.8 is considered a "large" effect; our values exceed this threshold by a factor of 3-4×, indicating a very substantial separation between the chosen and rejected centroids in activation space.
> These results will be included in the revised manuscript and confirm that the cluster structure in Figure 2 reflects a statistically robust phenomenon, not an artefact of visualisation.
>
> ## W3: Potential data contamination from pretraining
>
> This is an important concern. To control for this, we ran an additional experiment using Llama 2 (released July 2023) paired with the Nectar dataset (released November 2023), ensuring a strict temporal separation where the model could not have seen the data during pretraining. The results closely mirror those reported in the paper: peak probe (val) accuracy is 78.3% ± 1.9, t-test on PC1 yields p << 0.001, and a Cohen’s d of 3.73.
>
> This demonstrates that the activation-level preference structure is not an artefact of memorisation but reflects a genuine geometric property of how language models represent response quality.
>
> ## W4, Q1: Confidence intervals and class balance
> We confirm that the evaluation data is perfectly balanced (50/50 chosen vs. rejected).
>
> Confidence intervals: Across the same 9 settings tested in W2, we find probe accuracy confidence intervals ranging from ±1.7% to ±3.9% at most, with no systematic pattern across model families or datasets. These intervals will be added to Figure 3 and all accuracy tables in the revised version.
>
>
> ## W5, Q2: Original labels & equal-budget comparison
>
> We apologise for the ambiguity. In all main experiments (Tables 1, 4, 5), "Original labels" refers to training on the full ~50K labelled dataset, i.e., the strongest possible baseline. Our probe-labelled models use only 500 labelled examples for probe training, then propagate labels to the same 50K pairs. This means the comparisons in Table 1 pit 500 human annotations (ours) against 50K human annotations (original), making the competitive results of IPO, CPO, and KTO with probe labels all the more notable.
>
> Following the reviewer's suggestion, we also ran a head-to-head comparison where both the probe and the original-label baseline receive the same 500 labelled samples where the probe additionally uses the unlabelled 50K pairs through label propagation, while the original-label baseline trains only on the 500 examples directly. Using HH-RLHF (our main dataset) and DPO (deliberately chosen as the hardest setting for our method), the probe-labelled approach is the clear winner, with win-rate advantages (excluding ties) of up to +11.2% depending on the model. Only losing in the smallest setting of Qwen 0.6B.
>
> | Model | Probe wins | Original wins | Ties |
> | --- | --- | --- | --- |
> | Qwen 14B | **47%** | 42.8% | 24.40% |
> | Gemma 4B | **38%** | 37.6% | 24.40% |
> | Llama 3B | **41.2%** | 30.2% | 28.5% |
> | Llama 1B | **33.8%** | 30.4% | 35.8% |
> | Qwen 0.6B | 27.6% | **30.4%** | 42% |
>
> Here, we also want to draw attention to our Table 2, which reports the win-rates of the probe-trained model against the SFT baseline. Except for Qwen 14B, the win and tie ratios of this latest experiment closely resemble those against SFT, likely indicating that 500 samples are not enough for effective DPO training.
>
> This confirms that our pipeline's value lies precisely in its ability to amplify a small labelled seed into effective large-scale supervision.

---

> > ### Author Rebuttal · Reviewer_WTQ1 · 2026-04-01
> >
> > I wish to thank the author for addressing my concerns. While some of them have been addressed, I don't believe that all of them are resolved yet.
> >
> > **W1: Win-Tie-Loss representation and significance**: I strongly encourage the authors to use only a win-loss ratio, as this would make it clearer whether the method is effective. Otherwise, ties can just be seen as a proxy for uncertainty. While I see the author's point, one could also make the reverse argument: the tie-loss ratio of the reference model is much higher than that of the proposed method.
> >
> > **W2: Statistical Validation**: Thank you for doing this. I see this matter as resolved
> >
> > **W3: Potential Data contamination**: Thank you for doing this. I see this matter as resolved
> >
> > **W1, Q4**: Thanks. Is there any chance you could share the results with us (e.g., via an anonymous image-sharing link) so we can review and understand their significance?
> >
> > **W5** : Thank you for doing this. Apologies for being annoying, but I would again encourage to only use win-loss rates

---

> > > ### Author Response · Authors · 2026-04-07
> > >
> > > We thank the reviewer for their kind reply and for marking most issues as resolved.
> > >
> > > **W1: Win-Tie-Loss representation:** We replicate the main evaluations of Table 1 using a win-loss-only protocol (same judge, different prompt). Unfortunately, we were unable to integrate length-controlled judgments in our existing pipeline during the short duration of the discussion period but we will look at this option for the future.
> > >
> > > | Model | Baseline | DPO (Probe%) | IPO (Probe%) | CPO (Probe%) | KTO (Probe%) |
> > > |---|---|---|---|---|---|
> > > | Qwen 14B | vs Original | 27.20% | **82.60%** | 41.20% | 37.80% |
> > > |  | vs Random | 48.00% | **88.00%** | **51.80%** | **57.00%** |
> > > | Gemma 4B | vs Original | 27.20% | **61.00%** | 42.80% | 46.00% |
> > > |  | vs Random | 48.00% | 24.80% | **54.60%** | **97.00%** |
> > > | Llama 3B | vs Original | 31.20% | 44.60% | 28.60% | 43.00% |
> > > |  | vs Random | **55.60%** | 35.20% | 45.00% | **53.80%** |
> > > | Llama 1B | vs Original | 32.80% | **84.40%** | 36.80% | 37.20% |
> > > |  | vs Random | **52.60%** | **60.20%** | **54.80%** | **89.20%** |
> > > | Qwen 0.6B | vs Original | 26.40% | 48.60% | 41.00% | 48.40% |
> > > |  | vs Random | **55.40%** | **61.20%** | **52.60%** | **74.60%** |
> > >
> > > Bold marks cases where the probe-trained model wins outright. Compared to Table 1, the two protocols agree on 89% of probe wins.
> > >
> > > To understand the remaining 11%, we asked whether the disagreements stem from judge confidence (as the reviewer hypothesised) or from the stochasticity of generation. We picked two representative flips: Qwen 0.6B / DPO vs. Random (probe loses under win-tie-loss but wins under win-loss) and Gemma 4B / DPO vs. Random (the opposite). Repeating each evaluation 10 times yields 53.2 ± 2.1% probe wins for the former and 53.0 ± 2.3% for the latter; so the probe is in fact the meaningful winner in both cases, suggesting the original disagreement was driven by judge variance rather than a substantive flip.
> > >
> > >
> > > **W4, Q1:** Below are the full per-setting results we could not fit into the original rebuttal (CIs differ slightly from before due to random initialisation).
> > >
> > > | Model | Dataset | Pretrained acc ± std | PO acc ± std (%) |
> > > | --- | --- | --- | --- |
> > > | Gemma 4B | HH-RLHF | 58.2 ± 1.8 | 57.7 ± 3.2 |
> > > | Gemma 4B | Nectar | 77.7 ± 2.3 | 79.4 ± 1.5 |
> > > | Gemma 4B | UltraFeedback | 62.6 ± 3.3 | 62.8 ± 3.6 |
> > > | Llama 3B | HH-RLHF | 61.5 ± 1.9 | 61 ± 2.5 |
> > > | Llama 3B | Nectar | 79 ± 3.7 | 80.4 ± 3.3 |
> > > | Llama 3B | UltraFeedback | 60.2 ± 2.4 | 62.3 ± 1.8 |
> > > | Qwen 0.6B | HH-RLHF | 57.7 ± 1.6 | 59.4 ± 2.4 |
> > > | Qwen 0.6B | Nectar | 75.3 ± 4 | 76.3 ± 2.5 |
> > > | Qwen 0.6B | UltraFeedback | 62.1 ± 3.1 | 60 ± 3.1 |
> > >
> > >
> > > **W5:** Given the reviewer’s comment, we also replicate this new experiment with win-loss only, and report very similar results compared to win-tie-losses.
> > >
> > > | Model | Probe wins (%) |
> > > | --- | --- |
> > > | Qwen 14B | **52%** |
> > > | Gemma 4B | **51.4%** |
> > > | Llama 3B | **52.6%** |
> > > | Llama 1B | **50.8%** |
> > > | Qwen 0.6B | 44.4% |
> > >
> > > It is interesting to see the ties being ‘funnelled’ somewhat evenly among the two models. This makes both the wins of our method (4 out of 5) and the only loss (Qwen 0.6B) closer to the decision boundary.
> > >
> > > Here, we want to recall that this is already a stress test; we are using DPO (the method most sensitive to label noise) on HH-RLHF, so winning 4/5 settings under the stricter protocol still supports our central claim that probe-based label propagation amplifies a small seed effectively.
> > >
> > > ---
> > >
> > > We hope these additional results address any remaining concerns, and we thank the reviewer again for pushing us toward a stronger evaluation.

---

### Official Review · Reviewer_UCyP · 2026-03-12

**Soundness:** 3
**Presentation:** 3
**Significance:** 3
**Originality:** 3
**Overall Recommendation:** 5
**Confidence:** 4

**Summary:**

Summary: The paper proposes utilizing a linear probe trained on a few labelled preference pairs to annotate large unlabeled preference datasets. They demonstrate that activations from chosen and rejected responses form distinct clusters, but for preferences different from the ones used to train an SFT/PO model, they are less distinguishable, suggesting that a probe may be more appropriate for labelling in low-resource settings.

**Compliance With Llm Reviewing Policy:**

Affirmed.

**Final Justification:**

The paper provides a new method that allows for low resource preference adaptation that is well motivated, and they demonstrate that probes can capture the preference structure and that the method can provide benefits. In particular, during the rebuttal, they directly demonstrate that the method results in improvements on downstream tasks. While the method has some limitations, it demonstrates benefits and a comprehensive analysis, and it can open up new explorations of low-resource learning. Overall, I think the paper is impactful.

**Key Questions For Authors:**

- Would it be possible to provide some initial results on downstream preference optimization for PRISM? One possible evaluation is training a probe on a subset of users from a group from PRISM, annotating HH-RLHF or another dataset with the probe, and evaluating on the unseen users in PRISM.

**Limitations:**

Yes

**Strengths And Weaknesses:**

Strengths:

- The paper presents a novel method for annotating unlabeled datasets using a probe that is trained on a small number of samples.
- The paper performs an evaluation across methods, datasets, and models and demonstrates the benefits of the method as well as some analysis on the effect of robustness.
- The paper performs an analysis that demonstrates that probes can capture preference label structure and also why they would be useful for learning a set of preferences different from those in “standard” datasets.

Weaknesses:

- Evaluations on downstream preference optimization would be useful to directly demonstrate the applicability to the target setting. One possible evaluation is training a probe on a subset of users from a group from PRISM, annotating HH-RLHF or another dataset with the probe, and evaluating on the unseen users in PRISM.

---

> ### Author Rebuttal · Authors · 2026-03-30
>
> We thank Reviewer UCyP for the thorough and constructive review, and for recognising the novelty of our method, the breadth of our evaluation, and the analysis motivating why probes are particularly suited for learning preferences that diverge from standard datasets.
>
> **Downstream preference optimisation on PRISM**
>
> We appreciate the suggestion, and have conducted some preliminary experiments following a setup closely aligned with the reviewer's proposal. We select "18-24 years old - Africa" as a representative minority group and train the probe on 500 of its samples. We then use the probe to annotate the full PRISM dataset (~27K pairs) and run preference optimisation. As a baseline, we train using all available samples from that group (i.e., full supervision).
>
> Since we cannot access the original annotator population for human evaluation, we design an embedding-based evaluation protocol. For held-out samples from the target group, we generate completions from both (A) the probe-trained model and (B) the fully-supervised model. We compute vector embeddings of each model's generations using a separate embedding model (sentence-transformers/all-MiniLM-L6-v2), and compare their cosine similarity to a "ground truth" embedding derived from the original labelled completions of the target group.
>
> We report the improvement of the probe-trained model over the fully-supervised baseline (Δ = sim_probe − sim_original). Positive values indicate the probe-trained model's generations are closer to the target group's preferences:
>
> | Model | DPO | IPO | CPO | KTO |
> | --- | --- | --- | --- | --- |
> | Gemma 4B | -0.0057 (-3%)  | +0.0584 (+29%) | +0.0124 (+5%) | +0.0271 (+12%) |
> | Llama 1B | -0.001 (-0.5%) | +0.0061 (+3%)  | +0.0226 (+10%)  | +0.0131 (+6%) |
>
> In the majority of cases, the probe-trained model exhibits higher similarity to the target group's ground truth than the fully-supervised baseline, notably so for IPO, which shows up to 29% relative improvement. This is particularly striking given that the probe uses only 500 labelled examples while the baseline uses all available group-specific data. These results are also consistent with our main findings on standard datasets (Tables 1, 4, 5): IPO remains the most robust method under probe-generated labels, while DPO shows marginal degradation, further reinforcing our practical recommendation to prefer IPO in low-resource settings.
>
> We acknowledge that embedding-based similarity is an imperfect proxy for human preference judgement, and we discuss this transparently as a limitation. Nevertheless, we believe these results provide encouraging initial evidence that our pipeline transfers effectively to the culturally-specific, low-resource setting that motivates our work. We will incorporate these results and the evaluation protocol into the revised manuscript.

---

> > ### Author Rebuttal · Reviewer_UCyP · 2026-04-03
> >
> > Thank you for the response. The results provide a direct verification of the method's applicability to the target setting with noticeable improvement and show the insights generalize to the target setting as well.

---

### Official Review · Reviewer_xW9p · 2026-03-13

**Soundness:** 2
**Presentation:** 2
**Significance:** 2
**Originality:** 2
**Overall Recommendation:** 3
**Confidence:** 3

**Summary:**

This paper investigates how LLMs encode preference information within their intermediate activations, with finding suggesting that activations from chosen and rejected response pairs form clusters across layers. The Authors utilize this structure to propose training a lightweight linear probe on a few labelled preference pairs (≤500) and using then using it to annotate large unlabelled datasets for downstream preference alignment. Evaluations span several datasets, with model scales between 1B-14B. Results show models trained with probe-generated labels outperform SFT baselines and models trained on random labels. This work aims to democratise preference optimisation for low-resource settings, including underrepresented communities whose preferences have historically been underserved by large-scale models.

**Compliance With Llm Reviewing Policy:**

Affirmed.

**Final Justification:**

On the basis of the Authors' effort in the rebuttal, I have raised my score.  However I have remaining concerns that the small scale of the human feedback experiment is not indicative of performance. If it is intended as a side note, then more eval to backup performance is needed. "Table 1 alone covers 5 models × 4 PO methods × 2 baseline comparisons"  is considered small-scale compared to typical modern preference alignment research. More evaluation can help support the generalization/capabilities of this work.

**Key Questions For Authors:**

- Since probe accuracy peaks in middle-to-late layers, did you experiment with multi-layer ensembles or non-linear probes to see if they could better capture the preference signal than a single-layer linear probe?

- How does the diversity of the initial 500 samples affect the probe's generalization? For example, if the small seed set is biased toward a specific topic, does the probe fail to propagate labels accurately for other domains in the large dataset?

- Can you more clearly quantify the limitations and tradeoffs of the approach? Which scale of model it works best on, and at which scale does it fail to perform? Is there a specific domain this method is ideally suited for (such as creative writing, helpfulness-harmfulness, mathematical reasoning, etc) and can you give any qualitative examples of what culturally diverse preferences might be?

**Limitations:**

Yes.

**Strengths And Weaknesses:**

**Strengths**

- The problem is well motivated. Low-resource preference adaption is a relevant challenge as LLMs become more main-stream tools yet training data generally focuses on high volume domains. The method aims to achieve comparable performance gains with less data using 100x fewer human annotations when compared to traditional large-scale data labeling pipelines.

- The paper provides a comparison of preference optimization methods, revealing that IPO is highly robust to the noisy labels generated by the probe and DPO requires specific modifications like label smoothing to be effective.

- Training the linear probe is negligible (under a minute), and the label propagation process can be fused with the optimization training to eliminate extra costs. This is a highly computationally efficient feature of the work.

**Weaknesses**

- The method likely has limited performance/a performance ceiling. Probe accuracy is limited by the noise in human preferences. On the HH-RLHF dataset, probe accuracy plateaus at ~63%. Section 3.3 and Figure 3 almost demonstrate accuracy plateaus.

- The paper relies heavily on LLM-as-judge, which may introduce its own biases and is known to be noisy. As a result it is unclear if the method can effectively capture the personalized and cultural preferences it is intended to support.

- Small models (Qwen 3 0.6B) seem to degenerate easily regardless of whether it used original or probe-generated labels. It is unclear at what size of model the simple linear probe approach might start to yield performance gains/when it fails. This area lacks transparency and analysis. The intuition that the method relies on the underlying geometric structure of how LLMs process information is not mathematically well-defined or presented, therefore limitations of exactly when and why the method can be performant are not clear.

---

> ### Author Rebuttal · Authors · 2026-03-30
>
> We thank Reviewer xW9p for their thoughtful evaluation and for recognising the practical relevance of our method, the computational efficiency of the pipeline, and the systematic comparison across preference optimisation methods. We address each point below.
>
> > **How does the diversity of the initial 500 samples affect the probe's generalization?**
>
> To directly test how seed set composition affects generalisation, we conduct new cross-dataset transfer experiments using Llama 3.2 3B (pretrained), where a probe trained on one dataset is evaluated on held-out data from a different dataset:
>
> | Training ↓ / Test → | HH-RLHF | Nectar | Ultrafeedback |
> |---|---|---|---|
> | HH-RLHF | x | 55.2% | 50.8% |
> | Nectar | 50.7% | x | 55.2% |
> | Ultrafeedback | 51.5% | 60.2% | x |
>
> Roughly half of these settings approach random chance, indicating the probe learns dataset-specific preference structure rather than a universal quality signal. This is, in fact, a *desirable* property: it confirms the probe captures the particular preferences encoded in the seed set. When the seed set reflects a specific population's preferences, the probe propagates those preferences, not some other population's. However, this also implies that practitioners should ensure their seed set is representative of the target domain.
>
> > **Can you more clearly quantify the limitations and tradeoffs? Which scale works best, and at which scale does it fail?**
>
> Qwen 3 0.6B struggled most to learn stable preferences, but training with original labels on this model also frequently led to degeneration — suggesting the bottleneck is representational capacity, not probe label quality. The largest model (Qwen 3 14B) was most stable, which we attribute to richer representations yielding higher probe accuracy and more stable optimisation dynamics. Across the 1B-14B range, our method performs reliably when the base model itself can sustain preference optimisation.
>
> > **Is there a specific domain this method is ideally suited for?**
>
> Our method applies whenever preferences exhibit a binary or polar structure reflected in model activations, whether that is helpfulness, sentiment, toxicity, stylistic preferences, or any setting where responses can be meaningfully ranked. It is particularly suited to domains where (i) target preferences diverge from those in the model (making LLM-as-judge unreliable) and (ii) large-scale annotation is impractical.
>
> > **Can you give qualitative examples of culturally diverse preferences?**
>
> Research documents substantial cross-cultural variation: (i) *topic salience*: e.g., abortion is polarising in the US but less so elsewhere [2]; (ii) *perceived response quality*: country-level analyses reveal divergent model leaderboards [2]; and (iii) *preferred stance on sensitive topics*: attitudes toward religion, family structure, or political systems vary vastly [2]. For why diverse preference representation matters, we also refer to [3, 4], who document how alignment to a single population can systematically disadvantage others.
>
> > **The paper relies heavily on LLM-as-judge, which may introduce biases. Is it unclear if the method can capture personalized and cultural preferences?**
>
> We have conducted two additional experiments since submission. First, initial preference optimisation on the PRISM dataset shows mostly positive results consistent with our other findings (see our response to Reviewer UCyP). Second, a small-scale human evaluation corroborates the LLM judge rankings (see reply to Reviewer f91J). Our LLM judge results are also internally consistent across multiple judge models (Section 5.1). We will incorporate these results in the revision.
>
> > **Did you experiment with multi-layer ensembles or non-linear probes?**
>
> The probe is already trained on activations from three consecutive layers (Section 5.1), providing multi-layer aggregation within a single linear classifier. We deliberately favour simplicity: a linear probe confirms that preference information is linearly accessible in activation space, a stronger finding than showing it can be recovered by a more expressive model. Ensembling per-layer probes or using shallow MLPs are natural extensions we expect would yield incremental gains. We will discuss this in the revision.
>
> > **The intuition about geometric structure is not mathematically well-defined.**
>
> We agree that a formal treatment would strengthen the paper. Our contribution is primarily empirical: we document this structure across three model families, multiple scales, and diverse datasets, and show it can be exploited practically. Theoretical guarantees are an exciting direction which we leave for future work.
>
> **References**
>
> [2] Kirk et al., "The PRISM alignment dataset…" NeurIPS 2024
>
> [3] Kirk et al., "The benefits, risks and bounds of personalizing…" Nature Machine Intelligence, 2024
>
> [4] Ryan et al., "Unintended impacts of LLM alignment on global representation." ACL 2024

---

> > ### Author Rebuttal · Reviewer_xW9p · 2026-04-04
> >
> > Thank you for the Rebuttal, I have some additional questions please.
> >
> > - How will the above discussions be specifically integrated into the Revision?
> > - "We have conducted two additional experiments since submission" , Do you have any tables of results/plots that can demonstrate these two additional experiments? Also, small scale human experiments are not great signals or performance.
> > - On the final point, if the paper is presented as more empirically driven than theoretically backed, in the context of ICML, I believe some more discussion on the theoretical ideas will still be relevant.
> >
> > Thank you Authors.

---

> > > ### Author Response · Authors · 2026-04-07
> > >
> > > We thank Reviewer xW9p for engaging further with our rebuttal. We address each follow-up below.
> > >
> > > > How will the above discussions be specifically integrated into the Revision?
> > >
> > > We will use these discussions to improve the overall paper narrative. Specifically, we will expand the introduction with a discussion of how culturally-specific preferences manifest and why they matter, and we will add in both the introduction and the final discussion a characterisation of where our method is most suited, both in terms of model size and downstream task. The cross-dataset transfer experiment will be integrated into Section 3, and the multi-layer aggregation already used by our probe will be clarified in Section 5.
> > >
> > > In addition, we will also integrate improvements from the discussions with other reviewers.
> > >
> > > > Do you have any tables of results/plots that can demonstrate these two additional experiments?
> > >
> > > Yes, we provide the tables and additional details for the PRISM experiment in our reply to Reviewer UCyP, and for the human evaluation in our reply to Reviewer f91J. For ease of reading, we also add a copy here.
> > >
> > > Experiment on PRISM: the table reports the improvement of the probe-trained model over the 'original' baseline. A positive value indicates that the probe is better than 'original', indicating higher similarity between the generations and the ground truth.
> > >
> > > | Model | DPO | IPO | CPO | KTO |
> > > | --- | --- | --- | --- | --- |
> > > | Gemma 4B | -0.0057 (-3%) | +0.0584 (+29%) | +0.0124 (+5%) | +0.0271 (+12%) |
> > > | Llama 1B | -0.001 (-0.5%) | +0.0061 (+3%) | +0.0226 (+10%) | +0.0131 (+6%) |
> > >
> > > ---
> > >
> > > Human evaluation: this mimics our evaluation methodology, but with humans instead of LLM-as-a-judge. We report the Win/tie/loss rates of the probe-trained model vs the two baselines.
> > >
> > > | Model | Method | vs Original (W/T/L) | vs Random (W/T/L) |
> > > |---|---|---|---|
> > > | Gemma 4B | DPO | 21.7 / 17.4 / 60.9 | 42.0 / 30.0 / 28.0 |
> > > | Llama 3B | DPO | 20.9 / 14.0 / 65.1 | 28.1 / 43.8 / 28.1 |
> > > | Llama 3B | IPO | 76.2 / 4.8 / 19.0 | 52.6 / 21.1 / 26.3 |
> > >
> > > > Also, small scale human experiments are not great signals or performance.
> > >
> > > We fully agree that a large-scale human evaluation would offer stronger evidence. Unfortunately, the scale of our main experiment makes this infeasible within our resources: Table 1 alone covers 5 models × 4 PO methods × 2 baseline comparisons, which at 500 samples each would require 20K human judgments. We would also stress that this constraint is precisely the setting our method targets: scenarios where exhaustive human annotation is out of reach. Our small-scale human study is therefore intended as corroborating evidence for the LLM judge, not as a standalone validation, and it is consistent with the LLM judge rankings.
> > >
> > > > I believe some more discussion on the theoretical ideas will still be relevant.
> > >
> > > We appreciate this suggestion and agree that ICML readers would benefit from more theoretical grounding. In the revision, we plan to add a discussion connecting our empirical observations to existing theoretical literature. Specifically:
> > >
> > > The linear representation hypothesis posits that high-level concepts correspond to linear directions in LLM activation space. Jiang et al. [1] prove that the softmax cross-entropy objective, combined with gradient descent's implicit bias, mathematically promotes linear encoding of latent concepts. From the capacity side, Elhage et al. [2] show that sparse features are stored as linear directions via superposition, producing the geometric structure our probes exploit. Together, these results explain why preference information is accessible linearly in middle-to-late layers, and why this structure strengthens with SFT and preference optimisation.
> > >
> > > Probing theory [3, 4, 5] also establishes that linear probe accuracy lower-bounds the mutual information between a property and its representation, and that linear probes are substantially more selective than MLPs: their accuracy reflects what is encoded rather than what the probe itself can learn. This justifies our choice of a linear classifier as a diagnostic and practical tool: high accuracy is strong evidence of actual linear accessibility.
> > >
> > > ---
> > >
> > > We want to be transparent that providing formal convergence guarantees for preference optimisation under probe-generated noise remains an open problem, but we believe the discussion above would substantially strengthen the theoretical positioning of the paper while remaining honest about the boundaries of our current contribution.
> > >
> > > ### References
> > > [1] Jiang et al., On the Origins of Linear Representations in Large Language Models. ICML 2024.
> > >
> > > [2] Elhage et al., Toy Models of Superposition. Anthropic, 2022.
> > >
> > > [3] Pimentel et al., Information-Theoretic Probing for Linguistic Structure. ACL 2020.
> > >
> > > [4] Voita & Titov, Information-Theoretic Probing with Minimum Description Length. EMNLP 2020.
> > >
> > > [5] Hewitt & Liang, Designing and Interpreting Probes with Control Tasks. EMNLP 2019.

---

### Official Review · Reviewer_f91J · 2026-03-20

**Soundness:** 2
**Presentation:** 2
**Significance:** 2
**Originality:** 2
**Overall Recommendation:** 3
**Confidence:** 3

**Summary:**

Traditional preference optimization methods such as DPO, IPO etc. require large datasets of human comparisons which might be hard to obtain in low resource settings. It would also be hard to get correct comparisons from an LLM since this might be unreliable due to the setting. This paper tries to address this using activation based linear probing in order to create labels for a larger unlabelled dataset using a small labelled dataset. They find that activations of preferred and rejected responses (in the labelled data) form separate clusters because of which they use a linear probe to predict labels. Probe labelled data has been shown to perform empirically better than SFT and random labelled data. Performance with respect to the original data seems to be dependent on the technique used with IPO showing competitive performance while DPO seems to perform worse with an improvement when label smoothing is used.

**Compliance With Llm Reviewing Policy:**

Affirmed.

**Final Justification:**

I remain somewhat unconvinced about the robustness of the method due to its performance with DPO. While the results with other methods (e.g., IPO, CPO, KTO) are more encouraging, this suggests that the effectiveness of the approach may be dependent on the specific optimization technique. As such, I find the contribution interesting and potentially valuable, but not yet sufficiently general. Regarding generalization over architectures, the suggested heuristic for layer selection seems reasonable, but empirical validation on more diverse or substantially different architectures would make this claim more convincing.

In light of these clarifications, I am revising my score to a weak reject. While I see merit in the idea and would not strongly oppose acceptance, I believe additional evidence is needed to support the generality of the proposed approach across optimization methods and architectures.

**Key Questions For Authors:**

1. As noted by the authors, having only the LLM as a judge seems a very significant limitation given the motivation itself is that the low resource setting implies that preferences cannot be reliably labelled by LLMs themselves. Any human evaluation, even on a single dataset would make the pipeline more convincing and I would consider changing my evaluation.
2. A more insightful analysis of which layer to pick over different architectures would be helpful.

**Limitations:**

yes

**Strengths And Weaknesses:**

Strengths:
S1) Significance - Introduces activation-based label propagation using linear probes, which allows preference optimisation with few annotated examples, making it practically useful. The performance of IPO using this pipeline and comparison against random labels is empirically convincing. The finding that there are separate clusters for the activations on the chosen and rejected responses is significant.

S2)  Presentation - In particular, I find the practical guidelines useful since the user is walked through the entire pipeline and clearly guided on the choices to make.

Weakness:
W1) Significance - The paper is missing a key baseline: training directly on the small labeled dataset used for probe training. Without this, it is unclear whether the proposed pseudo-labeling approach provides any advantage over simply using the available labeled data.  Reliance on LLMs as judges for evaluation makes it unconvincing. The contributions are also all over the place and the idea that the user has to pick IPO over DPO suggests that this technique is not yet generalizable.

W2) Soundness - Table 3 seems to indicate that probe-generated labels do not yield good performance when used with DPO. Even with label smoothing, the probe-based model consistently underperforms the model trained on original labels, with win rates remaining substantially below loss rates. This undermines their main claim that they “demonstrate a practical method for preference adaptation requiring 100× fewer human annotations” since it’s not effective.

W3) Originality - Prior work[1] has already demonstrated that linear probes can be trained on hidden states to predict preference judgments. In contrast, this paper uses such probes for generating pseudo-labels to train downstream preference optimization methods. While this represents a different application of probes, it builds directly on the same underlying assumption and does not constitute a fundamentally new modeling approach. Further this paper is not cited despite essentially having very similar claims on probes for preference data.

W4) Presentation - The paper is not very well written. The introduction does not clearly outline the contributions upfront. Many of the main points appear only later in the discussion/conclusion, which can make it hard for readers to immediately understand the novelty and impact. The supplementary is incomplete.

1- Improving Preference Extraction In LLMs By Identifying Latent Knowledge Through Classifying Probes - ACL 2025

---

> ### Author Rebuttal · Authors · 2026-03-30
>
> We thank the reviewer for their detailed and constructive feedback. We address each point below and believe that the new experiments and clarifications substantively strengthen the paper.
>
> **W1: Missing baseline (training on small labelled set directly)**
>
> We conduct this new experiment. Comparing models trained on 500 probe-labelled vs. 500 original-labelled examples using DPO (our hardest setting), probe-labelled models win, with win-rates exceeding original-labelled models by up to +11.2% (excluding ties). Note that the "Original" baseline in the main paper used the full 50K dataset, a 100× annotation advantage. We refer to our reply to reviewer WTQ1 for more details.
>
> **W2: DPO performance with probe labels**
>
> The reviewer's concern focuses on DPO in isolation. While DPO with probe labels underperforms DPO with 50K original labels, this comparison spans a 100× supervision gap. Even so, DPO with probe labels consistently outperforms both SFT and random labelling (Tables 1, 2), confirming the probe captures genuine preference signal.
>
> The other three of our four methods (IPO, CPO, KTO) perform competitively or favourably with probe labels. IPO matches or exceeds fully-supervised performance across scales (e.g., 92.6% win+tie on Llama 3.2 1B, 86% on Qwen 3 14B). The central claim, "effective preference adaptation with 100× fewer annotations," should be considered in relation to the full result set.
>
> **W3: Relationship to [1]**
>
> We thank the reviewer for bringing this work to our attention. We were not aware of [1] during submission and agree it should be cited. We will add it in the revised related work section with an explicit discussion of the relationship.
>
> We would like to highlight several key differences that, in our view, make the two contributions distinct and complementary:
>
> - *Scope*: [1] evaluates probes only on post-trained models (i.e., models that have already undergone preference optimisation). Our method also applies to pretrained models, enabling a complete pipeline from pretrained checkpoint + 500 labels + unlabelled corpus to a completely aligned model.
> - *Core contribution*: [1] proposes probes as a replacement for LLM-as-judge in evaluation. We address whether probe-generated pseudo-labels can serve as training signal for preference optimisation, a question [1] does not study.
> - *Population-specific preferences*: We show in Section 3.5 that models already aligned to one set of preferences lose the ability to distinguish a different population's preferences in activation space (Figure 4). This finding directly challenges the assumption in [1] that post-trained models are the right substrate for probing, and motivates our use of pretrained models for culturally diverse settings.
> - *Resource regime*: We operate with 100-500 labelled examples vs. 5,000 in [1].
>
> **W4: Writing and presentation**
>
> We appreciate this feedback and will restructure the introduction in the revision to foreground the contributions more clearly. Specifically, we will: (i) move the key findings (method robustness differences, practical recommendations) into a concise contributions list at the end of the introduction; (ii) tighten the narrative arc so the reader immediately understands what is novel (probe-based label propagation for PO, not just probing per se); and (iii) complete the supplementary material with all missing details.
>
> **Q1: LLM-as-judge limitation and human evaluation**
>
> Following the reviewer's suggestion, we conducted a human evaluation on HH-RLHF with four annotators (non-authors), collecting 250 annotations across three configurations (Gemma 3 4B/DPO, Llama 3.2 3B/DPO, Llama 3.2 3B/IPO). Results (win/tie/loss for probe-labelled model):
>
> | Model | Method | vs Original (W/T/L) | vs Random (W/T/L) |
> |---|---|---|---|
> | Gemma 4B | DPO | 21.7/17.4/**60.9** | **42.0**/30.0/28.0 |
> | Llama 3B | DPO | 20.9/14.0/**65.1** | **28.1**/43.8/28.1 |
> | Llama 3B | IPO | **76.2**/4.8/19.0 | **52.6**/21.1/26.3 |
>
> We note that the relatively small sample sizes per configuration reflect the practical constraints of human evaluation, but the directional agreement with our LLM-judge results across all six comparisons (three configurations × two baselines) provides confidence that the automated evaluation faithfully reflects human preferences on this dataset
>
> Additionally, we have also conducted initial experiments on PRISM, yielding positive and encouraging results consistent with the activation geometry analysis in Section 3.5. We provide further details in our response to Reviewer UCyP.
>
> **Q2: Layer selection across architectures**
>
> Across Llama 3.2, Gemma 3, and Qwen 3, probe accuracy consistently peaks in layers ⌊L/2⌋ to ⌊2L/3⌋, with peaks shifting by at most 1-2 layers across families. For substantially different architectures, we recommend a quick sweep using ≤100 labelled examples, which completes in minutes.

---

> > ### Author Rebuttal · Reviewer_f91J · 2026-04-04
> >
> > Thank you for the detailed rebuttal. I find that concerns W1, W3, W4, and Q1 have been adequately addressed, particularly with the addition of the missing baseline and the inclusion of human evaluation.
> >
> > I remain somewhat unconvinced about the robustness of the method due to its performance with DPO. While the results with other methods (e.g., IPO, CPO, KTO) are more encouraging, this suggests that the effectiveness of the approach may be dependent on the specific optimization technique. As such, I find the contribution interesting and potentially valuable, but not yet sufficiently general.
> >
> > Regarding Q2, the suggested heuristic for layer selection seems reasonable, but empirical validation on more diverse or substantially different architectures would make this claim more convincing.
> >
> > In light of these clarifications, I am revising my score to a weak reject. While I see merit in the idea and would not strongly oppose acceptance, I believe additional evidence is needed to support the generality of the proposed approach across optimization methods and architectures.

---

### Decision · Program_Chairs · 2026-04-30

**Decision:**

Reject

**Comment:**

Summary: This paper investigates how LLMs encode preference information in their intermediate activations, finding that chosen and rejected response activations form distinct clusters even in pretrained models. Exploiting this structure, the authors propose training a linear probe on as few as 100–500 labelled preference pairs to pseudo-label large unlabelled datasets for downstream preference optimization. Systematic evaluation across three model families and four methods shows that models trained on probe-generated labels consistently outperform SFT and random-label baselines.

Strengths: Reviewers identified the problem setting of learning preferences with limited data is important, and highlighted the finding that preference information is linearly accessible in activation space was rigorously validated, and of broader interest. Reviewers also found the paper to be well-written and appreciated strong results on methods such as IPO.

Weaknesses: The primary weakness is that the results are not strong enough. Two reviewers pointed out that the method's effectiveness is strongly dependent on the choice of optimization algorithm, which was not resolved during rebuttal. IPO performs well with noisy probe labels, but DPO degrades substantially. This is problematic as DPO is a widely used method for preference learning, and the authors have not given sufficient reason why. Other concerns, for example regarding the win-tie-loss evaluation were addressed with a win-loss-only replication, but win rates are less than 50% for many methods outside IPO.

**Final Recommendation: Reject**

Justification: Reviewers were overall divided on the paper. The primary limitation highlighted by multiple reviewers that remains unresolved is that the method does not perform well with DPO, the most widely adopted preference optimization method. The authors ran some experiments, and addressed several concerns from the reviewers leading to some score increases.  However, two reviewers felt that the outstanding concerns such as applicability to multiple methods did not convince them. While the other reviewers did recommend the paper (even strongly), they still made a note of limitations.  I have chosen to focus on the limitations as the limitation in particular of performance on methods outside IPO is substantial. Win-tie ratio in the original portion of the Table 1 main results are less than 50% in many cases.  This is further reduces in the win-rate, highlighting improvements are smaller than initially stated.  Nonetheless, the paper presents an interesting and practical idea to use linear probes on LLM activations to propagate preference labels from small seed sets. Authors are encouraged to investigate performance for other preference optimization methods, especially DPO.